# A direct excitatory projection from entorhinal layer 6b neurons to the hippocampus contributes to spatial coding and memory

Yoav Ben-Simon [1,2] ✉, Karola Kaefer [1,3], Philipp Velicky [1], Jozsef Csicsvari [1], Johann G. Danzl [1] & Peter Jonas [1]

The mammalian hippocampal formation (HF) plays a key role in several higher brain functions, such as spatial coding, learning and memory. Its simple circuit architecture is often viewed as a trisynaptic loop, processing input originating from the superficial layers of the entorhinal cortex (EC) and sending it back to its deeper layers. Here, we show that excitatory neurons in layer 6b of the mouse EC project to all sub-regions comprising the HF and receive input from the CA1, thalamus and claustrum. Furthermore, their output is characterized by unique slow-decaying excitatory postsynaptic currents capable of driving plateau-like potentials in their postsynaptic targets. Optogenetic inhibition of the EC-6b pathway affects spatial coding in CA1 pyramidal neurons, while cell ablation impairs not only acquisition of new spatial memories, but also degradation of previously acquired ones. Our results provide evidence of a functional role for cortical layer 6b neurons in the adult brain.

The hippocampus is a cortical region in the mammalian brain required for mnemonic processes and spatial-related behavior[1]. Based on classical morphological work, the circuit of the hippocampal formation (HF) is often viewed as a loop, which begins at the two superficial layers 2 and 3 of the entorhinal cortex (EC) and ends in its deeper layers[2,3] after being relayed through the major hippocampal sub-regions—dentate gyrus (DG), CA3 and CA1—in a unidirectional fashion[4,5]. In light of this circuit architecture, it is surprising that various manipulations to the EC superficial layers only result in minor changes in hippocampal principal neurons activity and dynamics[6–11]. Thus, the presence of alternative synaptic pathways needs to be considered.

We recently developed a technique for reliable and efficient rabies-based transsynaptic retrograde labeling[12], which improves existing technologies[13,14]. By using this technique, we found that the hippocampus not only receives canonical input from layers 2 and 3 of the EC, but also from its deep layers, particularly layer 6[12]. As layer 6 constitutes a major part of the EC, and contains a large proportion of excitatory neurons, this brain region may have considerable impact on the activity of the hippocampal network. To date, neurons of this layer

have been categorized exclusively by their characteristic polymorphic dendritic morphology[2,3,15,16]. Although this layer was shown to house cells with spatially selective firing patterns, reminiscent of grid cell activity[17], no detailed molecular, anatomical, physiological, or functional characterization of this cell population has been performed.

A major difficulty in the analysis of the function of layer 6 is its complex organization. Although the cortical layer 6 is often considered a single uniform layer, neurons of layer 6b (sometimes referred to as layer 7[18]) are different from neurons in layer 6a developmentally, genetically and morphologically[19,20]. Neurons in layer 6b differentiate much earlier than neurons in layer 6a and express several markers specific to subplate neurons (SPNs), including Complexin 3 (*Cplx3*), Neurexophilin 4 (*Nxph4*), and the connective tissue growth factor (*Ctgf*)[21–23]. Although it is generally thought that SPNs are a transient population[24,25], the presence of a persistent fraction of SPNs has been demonstrated[22,26]. Notably, the entorhinal layer 6 is more akin to cortical layer 6b[27], rather than layer 6a[28], and often appears to be continuous with the thin cell layer thought to represent the remnant of the subplate[18,24,26]. These results raise the possibility that beyond their

[1]Institute of Science and Technology Austria (ISTA), Klosterneuburg, Austria. [2]Present address: Department of Neurophysiology and Pharmacology, Vienna Medical University, Vienna, Austria. [3]Present address: Department of Neuroinformatics, Radboud University, Nijmegen, The Netherlands. ✉e-mail: yoav.bensimon@meduniwien.ac.at

function in cortical development[24,29–31], layer 6b neurons may regulate circuit function in the adult hippocampus. However, this hypothesis has not been directly tested.

To study the connectivity and network function of EC deep layer input to the hippocampus, we combined rabies virus-based transsynaptic labeling, molecular characterization, electrophysiology, optogenetics, and behavioral analysis. We found that, in the adult brain, EC-6b neurons are monosynaptically connected to hippocampal principal neurons and have a surprisingly powerful effect on hippocampal network activity, from the single-neuron up to the behavioral level.

## Results

### A direct EC-6b excitatory hippocampal pathway

To attain a better molecular characterization of EC-6 neurons, we first examined the expression of several neuronal markers (Supplementary Fig. 1a–c). We found that this layer does not express known markers for the neocortical layer 6a, such as *Ntsr1*, but was rich in known biomarkers for SPNs, such as *Cplx3*, *Nxph4*, and *Ctgf*[21–23] (Supplementary Fig. 1a–c). Using a single-cell RNA sequencing database[32] and fluorescent in-situ hybridization (FISH), we found that in the adult posterior cortex, these hallmark SPN genes are powerful selection markers for a distinct population of excitatory neurons, unique both within the cortical layer 6 cluster, and within the cortical plate at large (Supplementary Fig. 1d–h). Consistent with these observations, immunolabeling for CPLX3 produced a strong signal in somata of cells in the deepest cortical layer, termed layer 6b or subplate (defined as a 50-μm wide band above the white matter[22]) but also along the cortical and hippocampal layer 1, i.e., *stratum moleculare* (*S.M.*), where only a few CPLX3⁺ somata could be detected (Fig. 1a and Supplementary Fig. 2a and b).

Since CPLX3 is a synaptic protein[33], we hypothesized that CPLX3 immunoreactivity along the *S.M.* could indicate the existence of synapses arising from the population of persistent SPNs. Using stimulated emission depletion (STED) imaging of the *S.M.* taken from different hippocampal and parahippocampal sub-regions, we were able to confirm that in these regions, CPLX3 colocalized extensively with the vesicular glutamate transporter 1 (VGluT1) and the vesicular GABA transporter (VGAT), respectively. These excitatory or inhibitory CPLX3⁺ terminals (eCPLX3/iCPLX3) were detected in similar proportions to one another, but in variable proportions across the different sub-regions of the HF (Fig. 1b, c and Supplementary Fig. 2c, d). A notable exception was observed in the CA3 *S.M.*, where eCPLX3 terminals accounted for nearly 15% of all glutamatergic terminals, whereas iCPLX3 terminals were less than 5% of all GABAergic ones.

To test whether these terminals arise from EC-6b neurons, we genetically targeted CA3 pyramidal neurons for transsynaptic retrograde labeling by first injecting the contralateral CA1 of the tdTomato cre reporter line Ai9, with retrogradely-transportable adeno-associated virus (retAAV)[34] expressing cre-recombinase, and the ipsilateral CA3 with adeno-associated virus (AAV) expressing a cre-dependent TVA-2A-N2cG cassette. A subsequent injection of envA-pseudotyped, G-deleted CVS-N2c rabies viral vectors (RVdG_envA-CVS-N2c) expressing EGFP[12,14], two weeks following the application of both AAVs, led to extensive and specific retrograde labeling from CA3 pyramidal neurons (Fig. 1d and Supplementary Fig. 3a, b). In agreement with previous knowledge[35], widespread EGFP signal could be observed along layer 2 of the medial and lateral EC (MEC and LEC, respectively) and consistent with our prediction, also in CPLX3⁺ EC-6b neurons (Fig. 1e, f). Since in the developing brain, the subplate layer might give rise to both excitatory, as well as inhibitory projection neurons[36], we performed an additional retrograde labeling experiment using GAD1-EGFP mice and found that the vast majority of projection neurons from EC-6b were GAD1-negative (Fig. 1h and Supplementary Fig. 3c, d). Intracellular electrophysiological recordings and subsequent imaging of biocytin-filled CA3-projecting neurons in EC-6b revealed a polymorphic

arborization of densely spiny dendrites (Fig. 1i), spontaneous excitatory postsynaptic currents (EPSCs), and moderate spiking activity in response to current injections (Fig. 1j and Supplementary Table 1). These observations suggest that neurons in the deepest layer of the EC are persistent SPNs, continuous with the cortical layer 6b, and confirm the existence of a direct excitatory projection to the *S.M.* of the hippocampal CA3. In contrast, the iCPLX3 terminals we have detected may arise from local *Cplx3*⁺ *S.M.* interneurons, matching the molecular profile of previously described layer 1 neurogliaform cells[37] (Supplementary Fig. 4a, d). Taken together, these results demonstrate the presence of a direct excitatory synaptic connection from EC-6b neurons to the hippocampus.

### EC-6b neurons primarily contact hippocampal pyramidal neurons

To evaluate the anatomical extent of CA3-projecting EC-6b neurons, and quantify their relative proportion to other CA3-projecting cortical populations, we dissected out, optically-cleared, and imaged the entire HF, using confocal microscopy, following transsynaptic retrograde labeling from the CA3 (Fig. 2a, b). The resulting analysis revealed that CA3-projecting neurons were found along the entire dorsoventral axis of the EC-6b with a substantially higher density near its ventral pole, opposite the LEC (Fig. 2c).

Since eCPLX3 terminals were also observed in the *S.M.* of all other hippocampal sub-regions, albeit in lower densities, we extended this approach to include retrograde labeling from the CA1, DG and hippocampal interneurons. To identify presynaptic partners of CA1 pyramidal neurons, these cells were targeted using injection of a cre-off AAV cassette, under control of the CaMKIIa promoter[12] into the CA1 region of KA1-cre mice, in which cre selectively expresses in the CA3 and DG[38] (Fig. 2d), thus preventing retrograde labeling from nearby CA3 and most CA2 neurons (Supplementary Fig. 5a, b). These experiments revealed that, in addition to the principal EC-3 and an EC-5a input, which has recently been described in more detail[39] (Supplementary Fig. 5c, d), CA1 neurons also received a substantial input from the dorsal pole of the EC-6b, opposite the MEC (Fig. 2d, e, j) which similar to the EC-6b projection to the CA3, consists predominantly of excitatory neurons (Supplementary Fig. 6a).

To identify presynaptic partners of dentate granule cells (DGCs), we injected cre-dependent AAVs into the DGC-specific Prox1-cre line (Fig. 2f). We found that this population received uniformly distributed, but sparse excitatory input from the EC-6b (Fig. 2f–j and Supplementary Fig. 6b). Finally, to determine whether the EC-6b projection might recruit feed-forward inhibition, in addition to excitation, we performed retrograde labeling using RVdG_envA-CVS-N2c-tdTomato from hippocampal INs, by means of AAV-mediated expression of a Flp-dependent cassette in the hippocampus of the IN-specific DLX5/6-FlpE line, crossed to an EGFP Flp reporter line[12] (Fig. 2h and Supplementary Fig. 6c). Retrogradely-labeled cells included principal neurons in the hippocampus and in superficial cortical layers, along with very sparse labeling of neurons in EC-6b (Fig. 2h–j and Supplementary Fig. 6d, e). This result suggests that EC-6b neurons recruit less feed-forward inhibition than EC-2/3 neurons, which potentially increases their relative contribution to the overall extrinsic excitatory drive to hippocampal pyramidal neurons.

To determine the relative contribution of excitatory inputs from EC-6b onto each of these hippocampal populations, we calculated for each mouse the fraction of retrogradely labeled neurons in layer 6b of the total labeled population across all EC layers (Fig. 2k). We found that this ratio tracked the relative eCPLX3/VGluT1 synaptic density estimates described in Fig. 1c. Both the ratios reported in Fig. 1c and the ones in Fig. 2k provided independent, yet highly similar estimates to the contribution of the EC-6b projection to the net cortical input, thereby giving additional support to these results. This cross-validation enables approximation of the relative contribution of this

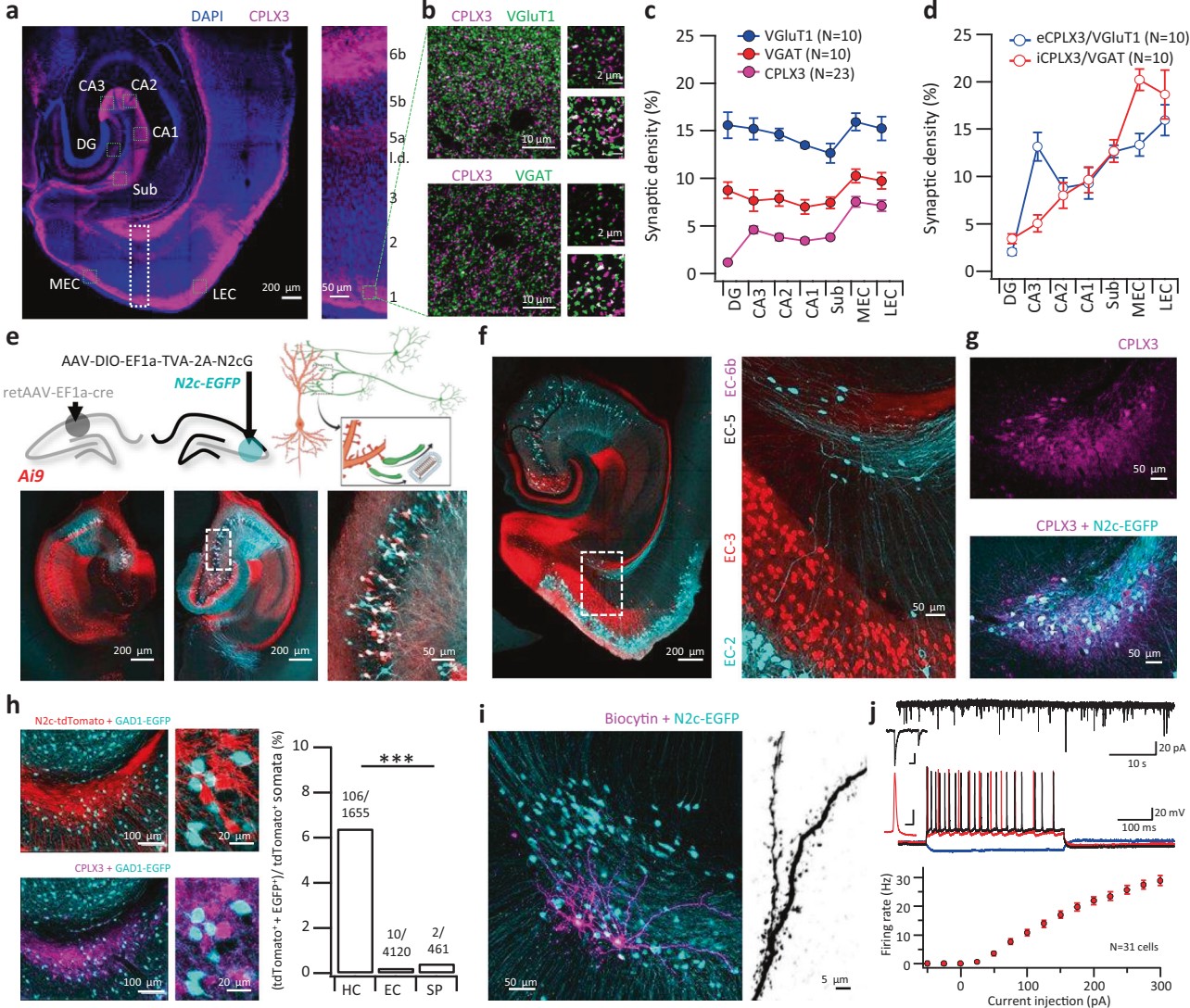

**Fig. 1 | A direct excitatory projection from EC-6b to hippocampal CA3.**
**a** Representative horizontal section stained for CPLX3 and DAPI (left) alongside an expanded view of the cortex (right). **b** Representative STED images from LEC-1, stained for CPLX3 alongside either VGluT1 (top left) or VGAT (bottom left). Expanded inserts are shown to the right of each image above their corresponding mask, demonstrating the calculation of colocalization for each marker. **c** Analysis of synaptic density, using STED images acquired from the fields denoted by green dashed squares in (a). Synaptic density was calculated here as the ratio between the signal area and total image area. **d** Relative eCPLX3 and iCPLX3 signal density of the total VGluT1 or VGAT signal area. **e** Unilateral injection of retAAV into the CA1 enables genetic targeting of the contralateral CA3 for retrograde labeling, exploiting the abundance of commissural axons of CA3 pyramidal neurons (top schematics). Representative horizontal sections from the mid-dorsal hippocampus of the ipsilateral (bottom left) and contralateral (bottom center) hemispheres for RetAAV-cre injection. Double-labeled neurons in the enlarged inset (bottom right) are putative starter cells. **f, g** Representative horizontal section from the ventral hippocampus following CA3-specific retrograde labeling reveals a population of

neurons in the deepest layer of the EC (**f**), which immunolabels positive for the subplate-specific marker CPLX3 (72/78 cells counted from 5 animals, **g**).
**h** Representative horizontal sections of the layer 6b following retrograde labeling with CVS-N2c(deltaG)-tdTomato, from CA3 neurons of a GAD1-EGFP mouse (left). Numbers above the bars in the right-hand panel indicate the total number of double-labeled tdTomato+/EGFP+ somata out of the total tdTomato+ somata per each of the regions examined ($P = 1.2 \times 10^{-9}$; two-sided Fisher's exact test). **i** Two biocytin-labeled CA3-projecting pSPNs (magenta) from an acute hippocampal section (left) and an enlargement of a dendritic segment (right). Experiment has been reproduced for 12 cells with identical results. **j** Representative traces of spontaneous EPSCs (top trace) and membrane potential following current injections of −50 pA (blue), 100 pA (red), and 200 pA (black; bottom traces). Summary plot below shows the firing frequency as a function of current amplitude ($N = 31$ cells). Scale bars for magnified representative EPSCs (top trace) and first AP at rheobase current (bottom trace) indicate 25 ms/20 pA and 2 ms/20 mV, respectively. $N$ in **c**, **d** represents number of individual sections from 6 different animals. Data in **c**, **d**, **j** is shown as mean and SEM.

projection into regions that cannot be easily dissected for retrograde labeling, using quantification of local eCPLX3 synaptic density. Additional quantification of the 2nd/1st order neuron ratio for the populations of excitatory neurons confirmed high-throughput labeling in all conditions, with lower ratios observed following retrograde labeling from the DG (Fig. 2l), potentially because of a high degree of divergence from a small number of EC neurons to a large number of DGCs[40]. These results demonstrate that the EC-6b projection targets all

hippocampal sub-populations, with preference toward excitatory neurons in the CA1 and CA3.

## The extended EC-6b network
Our results provide a detailed description of the EC-6b hippocampal projection. However, to understand what type of information these cells relay, a parallel functional characterization of its synaptic inputs is required. We found that the Ctgf-2A-dgCre driver line[41,42] enables

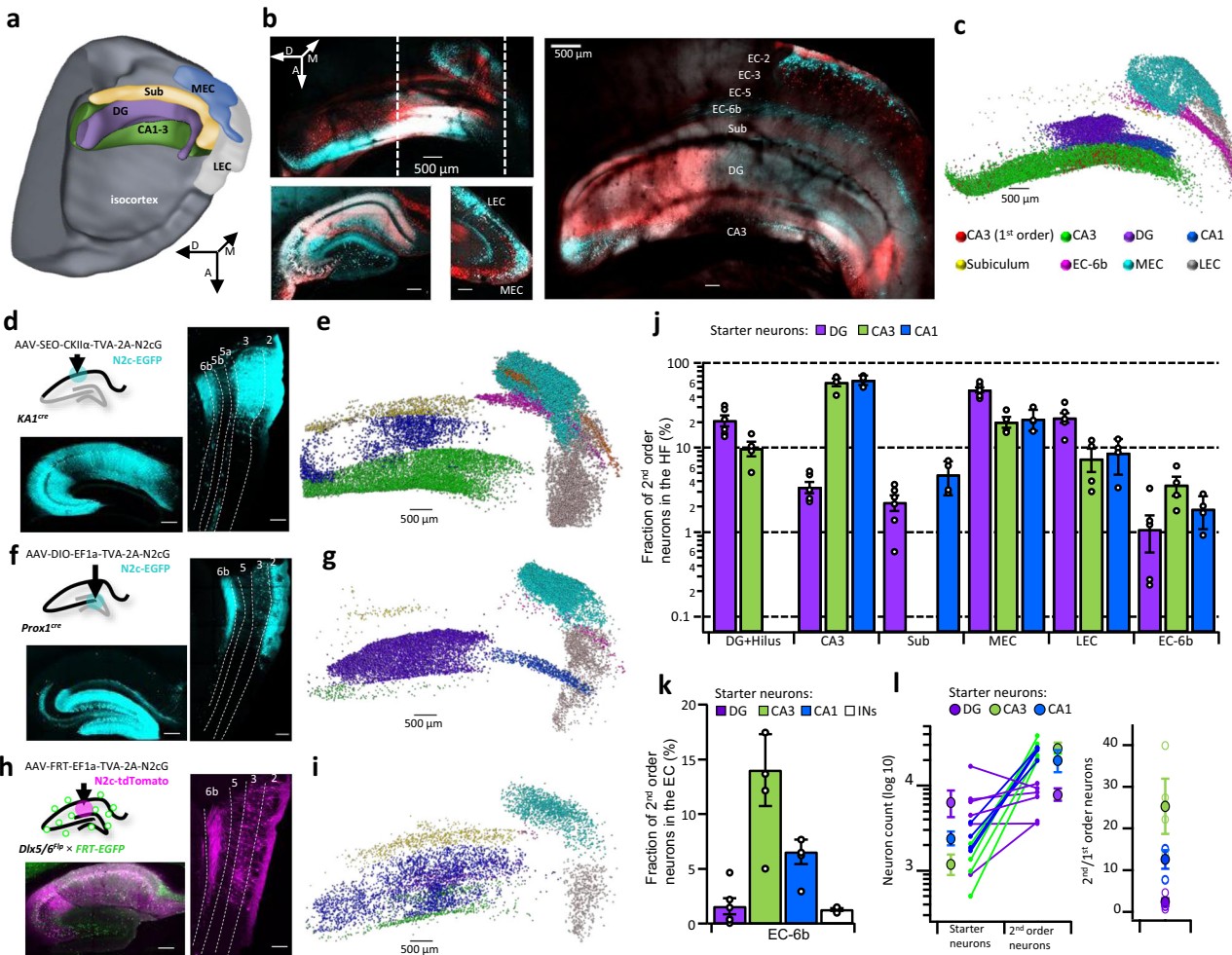

**Fig. 2 | EC-6b neurons project to all hippocampal sub-regions with a topographic organization. a** 3D schematic overview of the different components comprising the hippocampal formation. A, D, M indicate Anterior, Dorsal, and Medial, respectively. **b** Intact, cleared cortico-hippocampal preparation following retAAV-assisted labeling from CA3 neurons, as shown in Fig. 1e (red−tdTomato, Cyan−N2c-EGFP; top left), alongside projections of the XY plane along the traced lines (left, bottom) and an annotated image of the preparation in the Z plane (right). **c** Visual representation of the distribution of N2c-EGFP labeled neurons in the EC and HC, following retrograde labeling from the CA3. **d**, CA1-specific retrograde labeling scheme (top left) and representative confocal images of the HC (bottom left) and EC (right). **e** Same as **c** but for retrograde labeling from the CA1. **f** DGC-specific retrograde labeling scheme (top left) and representative confocal images of the HC (bottom left) and EC (right). **g** Same as **c** but for retrograde labeling from the DG. **h** CA1 IN-specific retrograde labeling scheme (top left) and representative

confocal images of the HC (bottom left) and EC (right). **i**, Same as **c** but for retrograde labeling from hippocampal interneurons. **j** Labeling distribution across cortico-hippocampal sub-regions following retrograde labeling from the CA3, CA1, and DG. Fraction in each region was calculated separately for each condition of the total number of 2nd order neurons. **k** Fraction of entorhinal SPNs out of the total number of projection neurons in the EC, following retrograde labeling from the CA3, CA1, DG, and hippocampal interneurons. **l** Numbers of 2nd and 1st order neurons for each starter population (left) along with a summary plot of the calculated ratios of 2nd to 1st order neurons (right). For all calculations, the number of CA3 and mossy cells was multiplied by 2 to account for the number of commissural projection neurons. For DG, $n = 6$, for all other conditions, $n = 4$ mice. Unless indicated otherwise, scale bars represent 200 μm. Image in **a** was acquired using the Allen Institute's Brain Explorer 2™. Data in j-l is shown as mean and SEM with individual data points shown for each experiment.

inducible genetic dissection of excitatory *Cplx3*+ EC-6b neurons, but not *Cplx3*+ EC layer 1 INs or hippocampal *S.M.* INs, upon administration of the antibiotic Trimethoprim (TMP) (Fig. 3a and Supplementary Fig. 7a–c). These mice were then used to target the EC-6b for dual anterograde and retrograde labeling, mediated by a combination of cre-dependent AAV and RVdG$_{envA}$-CVS-N2c vectors, in order to reconstruct their extended network of connections (Fig. 3b and Supplementary Fig. 7d, e).

Axon tracing from tdTomato+ neurons revealed that, apart from the structures within the HF, previously found to be immunoreactive for CPLX3, this population also sparsely projected to the lateral septal (LS) nucleus and the lamellar regions of the anterior thalamic nuclei (ATN). Concomitant retrograde tracing with RVdG$_{envA}$-CVS-N2c-EGFP revealed projections to EC-6b neurons from the medial septal nucleus (MS), and diagonal band of Broca (DBB), as well as the thalamic

core nuclei comprising the ATN (AD−Anterodorsal; AM−Anteromedial; AV−Anteroventral) along with the nucleus reuniens (Re) (Fig. 3d). From the cortical plate, sparse projections originated from the cingulate cortex (Cg), retrosplenial cortex (RSC) and the corticobasal amygdala (CoA). While all of these regions have previously been associated with the extended hippocampal network, we also found a prominent projection from the entire antero-posterior extent of the claustrum (Cla), whose association with the HF is rarely noted (Fig. 3c, d, f, g). EC-6b neurons also received extensive input from specific subpopulations within the HF; from the hippocampus proper, excitatory input arises exclusively from the CA1 and subiculum, along with a population of putatively inhibitory neurons in the CA3 *Stratum oriens* and *radiatum*. From within the EC, layer 5a emerged as the primary source of input to EC-6b, with sparser input from the other cortical layers, as distinguished by labeling for PCP4 which was found to be

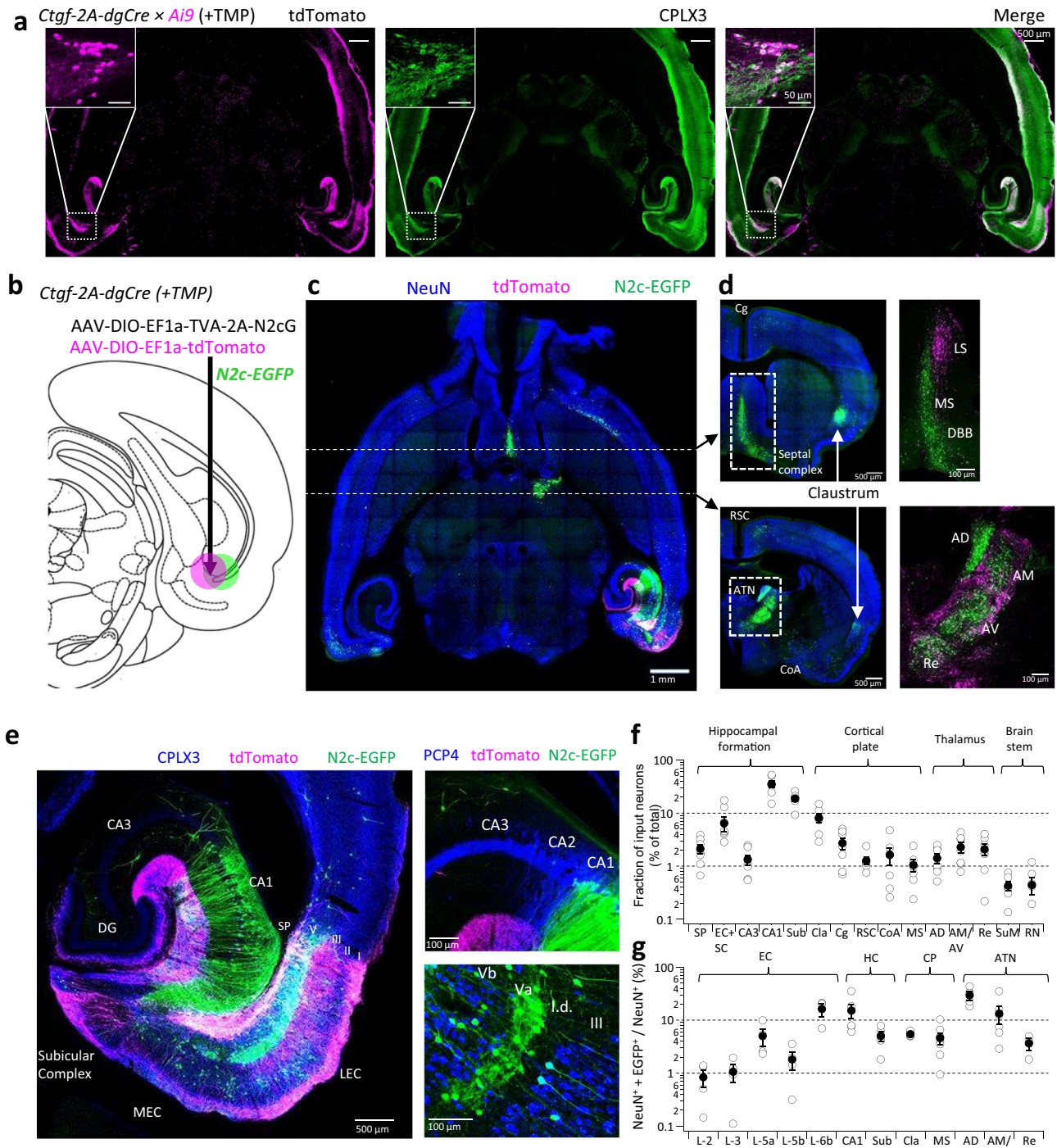

**Fig. 3 | Brain-wide connectivity of EC-6b neurons. a** Representative horizontal sections from Ctgf-2A-dgCre mouse crossed to the Ai9 tdTomato reporter line demonstrate efficient TMP-inducible genetic dissection of *Cplx3*+ neurons in the cortical layer 6b. **b** An illustration of the AAV injection scheme for subplate-specific dual anterograde and retrograde labeling. **c**, **d** A representative horizontal section following retrograde labeling from the entorhinal subplate and immunolabeling for NeuN (**c**) and representative coronal sections (**d**), demonstrating the projections to, and from, cortical and sub-cortical regions. **e** A representative horizontal section, immunolabeled for CPLX3 (left) and enhanced views of the CA3-CA1 transition area and the deep layers of the EC, following immunolabeling for PCP4. l.d.—*lamina dissecans*. **f** Summary plots showing the neuron count from all regions projecting to the entorhinal subplate, relative to the total number of EGFP+ neurons counted in each preparation, measured from intact and cleared cortical and brainstem tissue (*n* = 6 mice). **g**, Labeling density of NeuN+ + EGFP+ nuclei in key regions, relative to their respective total NeuN count (*n* = 5 mice). Data in **f**, **g** are shown as individual data points (open circles), mean (filled circles), and SEM.

selectively expressed in EC-3 and EC-5b neurons, but not EC-2 and EC-5a[43] (Fig. 3e–g).

## Synaptic physiology of EC-6b neurons

All the evidence presented so far points to the existence of a structural synaptic connection between EC-6b neurons and hippocampal principal neurons. However, it is still possible that these contacts are only structural remnants from early developmental stages and do not represent functional synapses in the adult brain. To address this possibility, we expressed an optogenetic actuator in layer 6b by crossing the Ctgf-2A-dgCre line with Ai32 mice, which conditionally express ChR2(H134R)-EYFP, and measured optogenetically-induced responses

from pyramidal neurons in CA3b (Fig. 4a and Supplementary Fig. 8a–c). To compare physiological properties of the EC-6b projection to the CA3 with other known pathways, we also expressed actuators specifically in EC-2 neurons and DGCs. The former was achieved by means of retrograde labeling from the DG, which lead to extensive labeling of EC-2, but only sparse labeling of EC-6b (Fig. 4b and Supplementary Fig. 8d–j), and the latter by means of AAV-EF1a-DIO-ChIEF-2A-dTomato injection into the DG of Prox1-cre mice (Fig. 4c). Whereas postsynaptic responses were observed following stimulation of the EC-6b pathway, these were markedly different from those of the perforant path (PP) or mossy-fiber (MF) projections to the CA3 (Fig. 4d–i). While the EPSC peak amplitude and latency to onset of EC-6b-CA3 EPSCs were identical to those of PP-CA3 EPSCs (Fig. 4j, k), the 20–80% rise time, decay time constant, and duration were significantly slower, and the short-term depression was nearly absolute (Fig. 4l–o). Similarly, EC-6b-CA3 excitatory postsynaptic potentials (EPSPs) have a slow time course, resembling previously described plateau potentials in hippocampal neurons[44] (Fig. 4p). Further recordings of optogenetically-evoked currents and potentials confirmed that EPSCs were sensitive to the AMPAR and KAR antagonist 6-cyano-7-nitroquinoxaline-2,3-dione (CNQX) under a negative holding potential ($-60$ mV) but that large NMDAR currents can still be observed under a positive holding potential ($+40$ mV). This provides additional confirmation that these currents are mediated predominantly by glutamate release and that the slow kinetics are not the result of polysynaptic activity, as this would be blocked by CNQX (Fig. 4q–r and Supplementary Fig. 9a–f). Taken together, these results indicate that EC-6b neurons establish a unique and powerful glutamatergic synaptic connection with CA3 pyramidal neurons.

## A role for EC-6b neurons in spatial information processing

While the existence of an excitatory projection capable of producing plateau-like postsynaptic potentials has not been predicted, similar activity patterns were previously shown to be sufficient for generation of new place fields in CA1 pyramidal neurons[44,45], suggesting that the EC-6b pathway might play a crucial role in spatial information (SI) processing. To test this hypothesis, we specifically expressed the optogenetic inhibitor ArchT in layer 6b neurons, by crossing the Ctgf-2A-dgCre driver line with Ai40 reporter mice, in which ArchT-EGFP is conditionally expressed (Fig. 5a). We then chronically implanted these animals with an optic fiber above the CA1 *S.M.*, along with six independently-movable tetrodes in the CA1 pyramidal layer (Supplementary Fig. 10a). We focused on the CA1 region, where we expected the largest effects of experimental manipulation, because direct effects via EC-6b-CA1 synapses and indirect network effects (EC-6b–DG–CA3–CA1 and EC-6b–CA3–CA1) might accumulate. Animals were then placed in a square arena, and following a 30 min long familiarization session (S1), left to continue exploration for an additional 30 min with photoinhibition activated only in one, randomly assigned quadrant (S2). This was followed by a third 30 min session, during which again, no photoinhibition was applied (S3) (Fig. 5b). Subsequent analysis of the SI score of putative pyramidal cells revealed that photoinhibition led to an acute decrease in SI only in the manipulated quadrant during the photoinhibition session, which was largely restored once photoinhibition was removed (target quadrant: $P = 1.02 \times 10^{-23}$; rest of arena: $P = 0.01$; Two-sided Mann–Whitney test, Fig. 5c). This decrease in SI could not be explained by any observable behavioral correlate, such as occupancy or movement speed, and similarly was not associated with any change in the basic firing properties of either excitatory or inhibitory neurons in the CA1 pyramidal cell layer (Supplementary Fig. 10b–j). As among all hippocampal-projecting neurons, only EC layer 6b expressed *Ctgf* (Supplementary Fig. 8a–c), these results indicate that EC-6b neurons regulate spatial coding.

## A role for EC-6b neurons in spatial learning and memory

Based on these findings, we hypothesized that this new synaptic pathway might be equally important in mediating formation and retention of spatial memory. To address this question, we crossed Ctgf-2A-dgCre mice with RosaStopDTA mice, to enable a selective conditional ablation of the EC-6b population in the adult brain, while sparing all other long-range excitatory hippocampal projections (Fig. 5d and Supplementary Fig. 8a–c). Double-transgenic animals were then co-housed in an intellicage environment[46], which allowed us to remotely control the location of water availability for each animal individually, and monitor its exploration behavior. Following a habituation period, we assigned each animal with a specific port (Fig. 5e; "A") where it could have unrestricted access to water and 5 days later, injected animals with either TMP, to induce specific ablation of layer 6b, or with vehicle solution. Immediately following this manipulation, we assigned each animal to a different water port, symmetrically rotated to a different corner and side for a period of 3 days. In total, this process was repeated three times (Fig. 5e; "B", "C", and "D") before reassigning the animals to the original port, whose location they learned prior to the manipulation (Fig. 5e, "A*"). We found that the ability of EC-6b-ablated animals to learn new reward locations became impaired ("D" versus "A"), at a rate which closely followed that of the ablation process (Fig. 5d). Unexpectedly, these mice also encountered difficulty forgetting the location they had learned immediately prior to the manipulation ("A*" versus "A").

Aligned rank transform three-way ANOVA analysis revealed significant effects of session and experimental day, and further demonstrated a highly significant interaction between group (TMP versus vehicle) and session (A, D, A*; $P = 1.3 \times 10^{-4}$), indicating that cell ablation significantly but differentially affected learning in novel and familiar sessions (Fig. 5e). In both novel session "D" and familiar session "A*", EC-6b-ablated animals showed reduced learning ("D": TMP, $P = 0.036$ versus Veh, $P = 1.8 \times 10^{-4}$; "A*": TMP, $P = 0.116$ versus Veh, $P = 2.2 \times 10^{-4}$). Since the effects in individual animals were variable, we hypothesized that the animals' performance will correlate with the degree of ablation. To test this hypothesis, we plotted the individual error rate on days 8 and 9 of the behavioral test, corresponding to the last day of the third novel port and the first day after return to the familiar port, against the degree of layer 6b ablation, measured by the density of the remaining eCPLX3 terminals relative to the total VGluT1 density in the CA3 (Fig. 5g). We found that acquisition of new spatial memories was significantly correlated with eCPLX3 density, whereas for retention of previously acquired memories there was a trend in the opposite direction (Fig. 5g; $r_s = -0.91$, $P = 3 \times 10^{-5}$ versus $r_s = 0.46$, $P = 0.11$). Thus, the animals with the most severe ablation were the ones with the poorest performance on day 8, and the best performance on day 9, and vice versa. These results confirm the involvement of EC-6b in both formation of novel and retention of old spatial memories.

## Discussion

### EC-6b neurons are putatively persistent subplate neurons

Our results demonstrate that neurons in the deepest entorhinal layer, matching the molecular[21–23], morphological[26,27], and electrophysiological[47] profile of cortical layer 6b, project to all hippocampal, and likely parahippocampal sub-regions, while exhibiting the characteristic bidirectional connectivity pattern with thalamic nuclei, previously reported for the SPNs in the neonatal brain[26,48–50] (Figs. 1–3 and Supplementary Figs. 1, 2). Thus, multiple lines of evidence suggest that the hippocampus-projecting EC-6b neurons we identified in the adult brain correspond to a persistent population of SPNs. This is an unexpected finding, not only because the existence of such cells in the EC was not previously reported, but more so because the deeper layers are thought to selectively process and redistribute the hippocampal output[2,3].

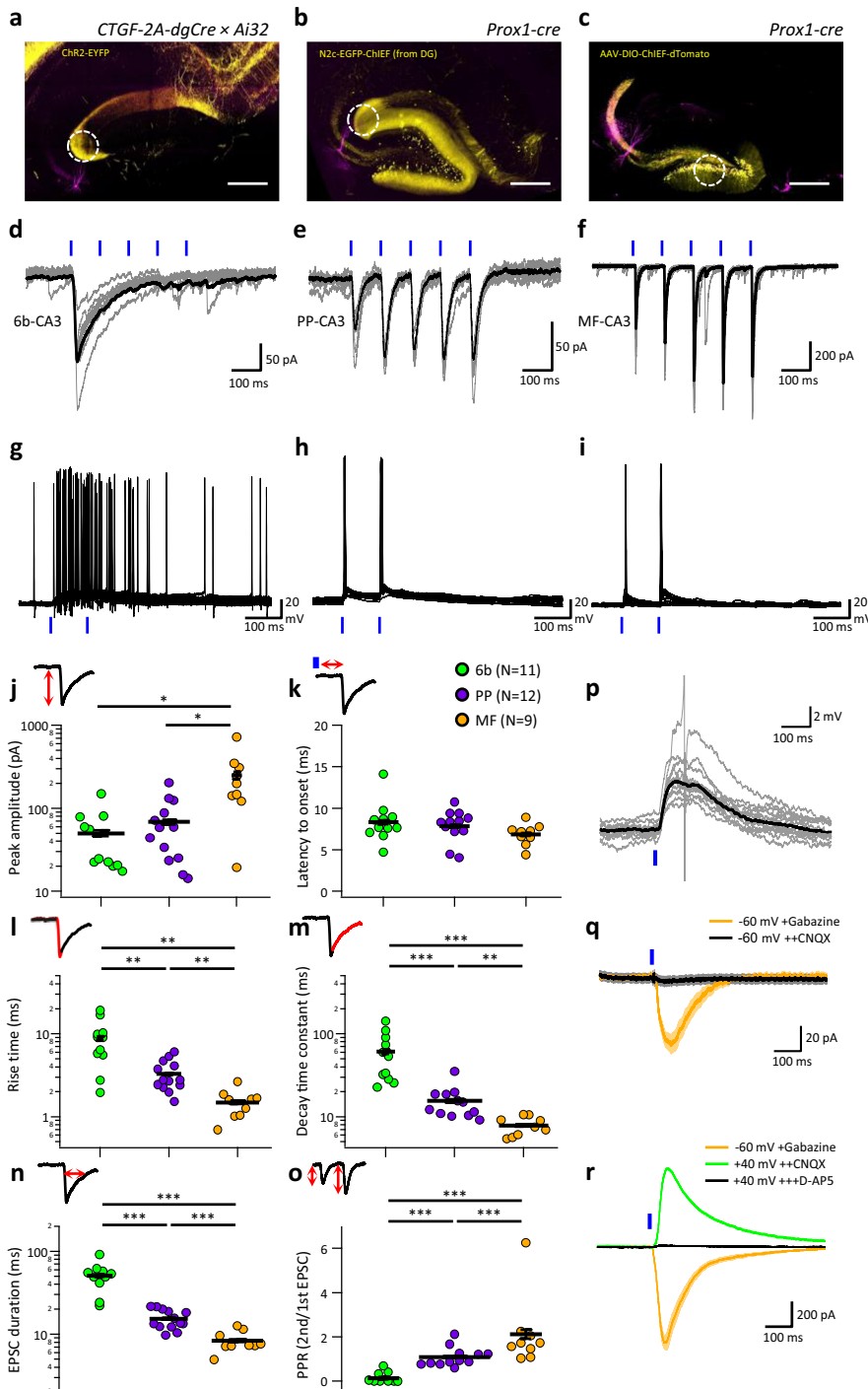

**Fig. 4 | Synaptic physiology of the EC-6b to CA3 projection. a–c** Representative images of acute hippocampal slices following specific expression of optogenetic actuators (yellow) specifically in EC-6b (**a**), EC-2 (**b**), or the DG (**c**) while recording post-synaptic responses from neurons in CA3b (magenta). White dashed circle indicates region targeted for photostimulation, and scale bars represent 200 μm. **d–f** Representative individual (gray) and averaged (black) traces of optogenetically-evoked EPSCs, following a 10-Hz train of 5-ms light pulses of the 6b-CA3 (**d**), PP-CA3 (**e**), or MF-CA3 (**f**) projections. **g–i** Ten overlaid voltage responses to 6b-CA3 (**g**), PP-CA3 (**h**), or MF-CA3 (**i**) stimulation. **j–o** Summary plots comparing the EPSC properties for each of the pathways: peak amplitude of the first response (**j**), latency to EPSC onset from the start of the stimulation (**k**), 20–80% rise time (**l**), decay time constant (**m**), duration at half maximum (**n**), and paired-pulse ratio at 100 ms interstimulus interval (**o**). Individual data points are shown as open circles and bars denote the mean and SEM for each condition. Schematics above the plots illustrate the measured EPSC property shown in each. **p–r** Representative 6b-CA3 responses showing plateau-like EPSPs with a single truncated AP (**p**; gray, individual traces; black, average), EPSCs recorded in the presence of 100 μM gabazine and 25 μM CNQX at a negative holding potential (**q**) and EPSCs recorded in the presence of 100 μM gabazine, 25 μM CNQX, and 50 μM D-AP5 at a positive holding potential (**r**). Number of measurements from each condition shown in parentheses in the legend (see **k**). Plots **j–o** show individual data points, with mean and SEM shown as vertical and horizontal black lines. Statistical differences with $P < 0.05$ using two-sided Mann–Whitney test (Holm–Bonferroni adjusted) were considered significant. Single (*), double asterisks (**), and triple asterisks (***) indicate $P < 0.05$, $P < 0.01$, and $P < 0.001$, respectively.

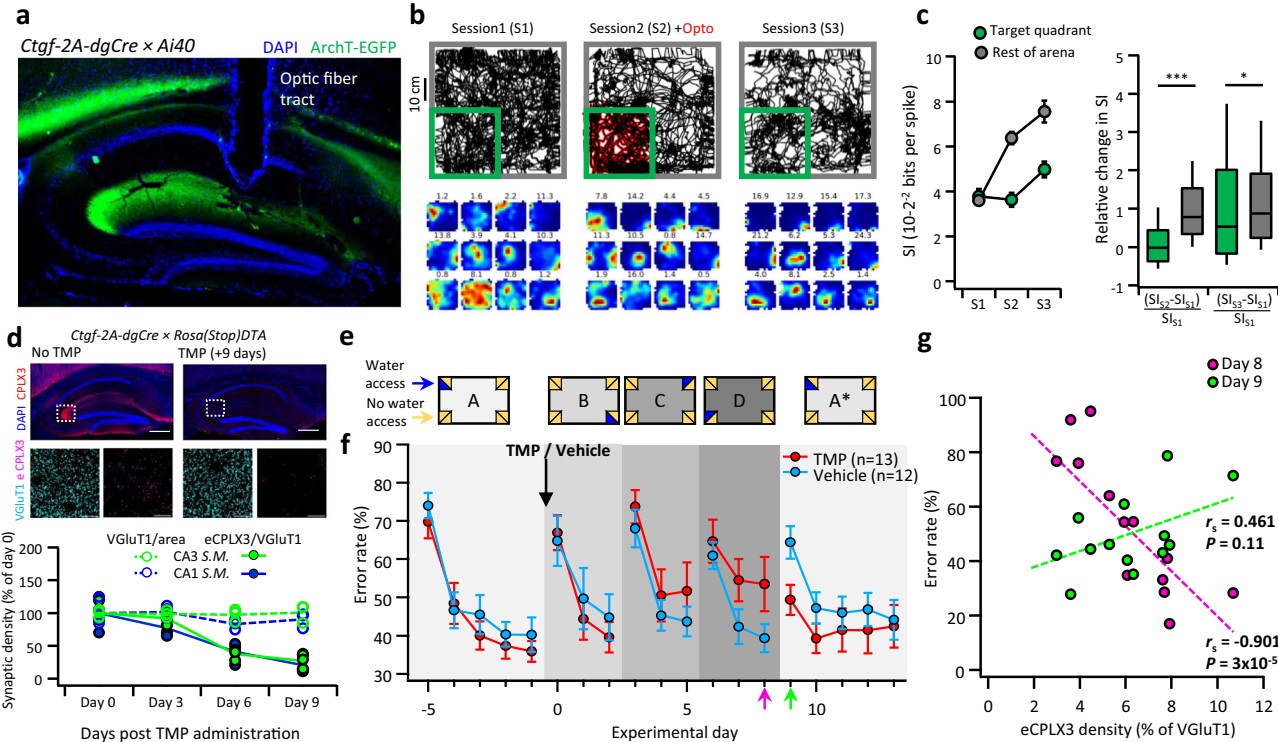

**Fig. 5 | Inhibition of EC-6b neurons affects spatial information processing and memory formation. a** Representative parasagittal section showing specific expression of ArchT-EGFP in layer 6b and the location of the optic fiber in a Ctgf-2A-dgCre crossed with an ArchT-EGFP (Ai40) cre reporter mouse. **b** Representative position plots for the three consecutive exposure sessions (top) and corresponding firing-rate maps of 12 representative units, ranked by their spatial information score in S3. Numbers above each plot note the peak firing rate (Hz). **c** A summary plot showing the average spatial information for CA1 pyramidal cells within the illuminated quadrant (green) relative to the rest of the arena (gray) in each exposure session (left) alongside a box plot showing the median change in SI between S2-S1 and S3-S1 for cells inside and outside of the target region (right). Boxplot center, bounds of box and whiskers represent the median, top and bottom 25th and top and bottom 10th percentile, respectively. A total of 184 units from 4 animals were included in the analysis. Single (*) and triple asterisks (***) indicate $P < 0.05$ and $P < 0.001$, respectively. **d** Representative images of the hippocampus of a Ctgf-2A-dgCre crossed with a Rosa(stop)DTA mice, immunolabeled for CPLX3, either without (top left) or 9 days following administration of TMP (top right) with representative STED images taken from the CA3 S.M. below demonstrate the density of VGluT1 and CPLX3 at each time point. Changes in density of both VGluT1 and eCPLX3/VGluT1 are quantified for CA1 and CA3 S.M. at different time points following administration of TMP, with each dot representing an individual animal ($n = 3$ animals per group) and lines represent the average for each condition (bottom plot). **e** Schematic diagram of the experimental protocol in the intellicage. **f** Time plot of the average group error rate during each of the experimental days. TMP was administered on experimental day 0, indicated by a black arrow. **g** Scatter plot between the density of surviving eCPLX3 terminals in the CA3 S.M., calculated from STED images, and the error rate for each animal in the TMP group, on experimental days 8 and 9, corresponding to the last day of the final novel location (D) and the first day of the return to the familiar location (A*), respectively. One-tailed Spearman's rank correlation coefficient ($r_s$) with its corresponding $P$ value indicated below and dashed regression lines shown for each group. Data in (**c**, left) and **f** is shown as mean and SEM and scale bars on images represent 200 μm.

SPNs are the first neurons generated during embryonic development, at E11.5–E12.5[29] and it is widely accepted that they have a prominent role in various processes associated with cortical development, such as cell migration, differentiation, and wiring of the cortical layers[24,51–53]. Consistent with this proposed function, it is widely assumed that SPNs die shortly after corticogenesis is completed, presumably by programmed cell death[24,25,54]. However, there is accumulating evidence that subpopulations of SPNs, preferentially those expressing *Cplx3* and *Ctgf*, persist into adulthood[18,22,26,55], and our results confirm and extend these previous findings. The contribution of SPNs to cortical development and of layer 6b to circuit dynamics in the adult may not be mutually exclusive and it will be interesting to examine possible interactions between the two processes in the future.

### A novel cortico-hippocampal pathway

Since the classical work by Santiago Ramón y Cajal[56] the EC-hippocampus circuit is often viewed as a trisynaptic loop, in which the major circuit elements are linearly connected. While minor additions were made, the basic connectivity scheme has remained unchallenged[5]. Our results identify a novel connection in this well-established circuit. Although the number of EC-6b neurons is much lower than that of EC-2/3 neurons, unique functional and structural

properties converge to endow these neurons with influence disproportionate to their numbers. First, the EPSC shows a slow decay and absolute synaptic depression, leading to plateau-like EPSPs in postsynaptic target cells, whose onset is timed to the first spiking event in a given burst (Fig. 4d, p). Second, EC-6b neurons receive convergent input from multiple sources, most notably the claustrum–a central and highly interconnected cortical hub[57] (Fig. 3), and target all sub-regions comprising the HF (Fig. 1d). Last, a high degree of inter-connectivity between SPNs, mediated by both chemical and electrical synapses, observed in the developing cortex[53,58,59] might persist into adulthood, allowing a small number of interconnected SPNs to synchronize activity across cortico-hippocampal sub-regions. Taken together, these unique properties may explain the powerful role of these neurons in behavior and hint at a uniformly important function across the cortical plate.

Several mechanisms may explain the slow decay of EC-6b-CA3 EPSCs. It is possible that glutamate receptor kinetics contributes to the slow time course, although the different kinetics of EC-2 and EC-6b inputs, which terminate on the distal dendrites of the same cell types, may argue against this possibility. Alternatively, asynchronous transmitter release, transmitter spillover and volume transmission could play a role. It is interesting to note that EC-6b neurons show similarities

to GABAergic *Ndnf*⁺ neurogliaform cells[37], notably regarding the slow time course of synaptic events[60] and the expression of the synaptic protein CPLX3, which is not expressed in any other cortical cell type (Supplementary Fig. 4). More work is needed to compare the two types of synapses at the unitary level.

### Role in spatial coding and memory formation

Our results suggest that EC-6b contributes to spatial coding and memory formation in the hippocampus. Acute optogenetic silencing of deep EC-6b neurons led to an acute decrease in SI of CA1 pyramidal neurons (Fig. 5a–c), with only minimal reduction in the average firing frequency (Supplementary Fig. 10). Thus, the effects of optogenetic inhibition differ from those observed following acute or chronic manipulation of superficial layer EC neurons, which have as yet resulted in only minor changes in SI[6–11]. Furthermore, the effects also differ from those of acute optogenetic inhibition of the CA3-CA1 projection, which lead to a marked decrease in firing rates, but an increase in SI in CA1 pyramidal neurons[61]. It is possible that the divergent output from layer 6b neurons affects the coherence of spiking in the hippocampal network, contributing to synchronization of principal neurons in the slow delta (0.5–3 Hz) and theta (3–8 Hz) frequency range. According to this model, place-specific firing patterns could arise and be stabilized as a result of a weak, but highly orchestrated activation of networks, spanning all hippocampal sub-regions. In the CA1, plateau potentials such as the ones we observed following stimulation of the EC-6b to CA3 projection, have been shown to underlie formation of place fields[44,45] and more recently, associative memories[62], providing additional support to this hypothesis. Multi-unit recordings from EC-6b previously showed that subsets of cells showed grid-like firing patterns[17]. Thus, the layer 6b connection to the hippocampus may provide an explanation for previous observations that hippocampal place fields can arise prior to, and in the absence of, EC-2 grid cell activity[8,63,64].

To investigate the contribution of EC-6b neurons to spatial memory formation, we designed an experiment using the intellicage system, which allows us to simultaneously and differentially control the location of a water reward for a cohort of co-housed animals, within their home cage and without the physical presence of an experimenter. This design, which somewhat mimics the situation in a natural habitat, revealed a dual effect of layer 6b ablation on spatial memory: while the ablation impaired the ability to learn new reward locations, it also affected the ability to forget previous locations.

The manipulation used to target EC-6b neurons was efficient in maximizing the number of affected cells in this layer, thereby allowing us to observe robust effects in a behavioral task. However, since the manipulation was not EC-specific, ablation of layer 6b neurons in other cortical areas may have also been involved. Although motor or sensory deficits can be ruled out, because they would result in a location-independent decrease in behavioral performance, more complex network effects beyond the EC cannot be excluded. These could arise from alterations to activity in the prefrontal cortex, which was previously shown to be involved in spatial tasks requiring cognitive flexibility[65,66] (but also see ref. 67). However, as the behavioral task has a major spatial component, which likely involves the EC–hippocampus axis, the most parsimonious explanation is that entorhinal layer 6b neurons play a critical role in these behavioral effects. Even in the unlikely scenario where the observed behavioral effects will be shown to require the prefrontal cortex only, with no involvement of the hippocampal formation, our findings should still hold relevance since, to-date, no behavioral or cognitive function has been assigned to cells of layer 6b in any cortical region.

Although this bidirectional effect might seem paradoxical, it has been suggested that memory degradation is an active and equally adaptive mnemonic process as memory encoding and that a reduced ability to forget should also be considered a memory deficit[68]. These findings raise the interesting possibility that these two opposing processes are not only related but arise from activity patterns of a single population of neurons, capable both of synchronizing, but also desynchronizing activity in hippocampal circuits.

We conclude that entorhinal layer 6b neurons are central to the proper function of the adult rodent hippocampus, including spatial coding and memory. Layer 6b neurons in other cortical regions may have similar functions, implying a general role in complex, brain-wide network computations. As the subplate layer is highly conserved across mammalian species and appears to be most developed in the human brain[69–71], our findings could potentially be extrapolated beyond the model organism. Future work will identify the contribution of layer 6b neurons to higher circuit functions of the human brain in health and disease.

## Methods

### Production of rabies viral vectors

HEK293-GT and BHK-eT cells were used to rescue, pseudotype, and amplify RVdG-CVS-N2c rabies viral vectors based on a previously published approach[12]. Briefly, HEK293-GT cells stably expressing the SAD B19 optimized glycoprotein and an optimized T7 RNA polymerase were transfected with the RVdG-CVS-N2c vector plasmid along with the SADB19 helper plasmids pTIT-N, pTIT-P, and pTIT-L using poly-ethylenimine (PEI). Once all cells in the culture were fluorescently labeled, usually 5–6 days from time of transfection and 1–2 days from the point fluorescence was first detected, the medium was harvested, filtered, aliquoted, and a small amount (100–200 μl) was transferred to dishes containing BHK-eT cells, which stably express the envA glycoprotein and the TVA receptor, for simultaneous pseudotyping and amplification of the vectors. Pseudotyped vectors were collected daily during a three-day period, starting at day 3 of the transduction and the virus was pooled and centrifuged at $70,000 \times g$ for 1.5 h. Following centrifugation, the medium was aspirated and the viral pellet resuspended in 200 μl phosphate-buffered saline (PBS), pH 7.4, aliquoted, and stored at −80 °C until use. Final titer of pseudotyped RVdG-CVS-N2c rabies viral vectors was determined using serial transduction of HEK293-TVA and calculated using a previously published formula[72]. For a full list of RVdG-CVS-N2c plasmids used in this study, see Supplementary Table 2.

### Production of adeno-associated viral vectors

AAV production was performed in HEK293T cells based on a previously published protocol[73]. Briefly, fully confluent HEK293 cells were transfected with an AAV2 vector plasmid along with pAdenoHelper and the AAV-dj RepCap plasmids using PEI. Thirty-six hours post transfection, the cells were harvested, pelleted, and lysed using three freeze-thaw cycles. The lysed cells were incubated with benzonase-nuclease (Sigma-Aldrich) for 1 h and then the debris was pelleted and the virus-containing supernatant collected and passed through a 0.22-μm filter. The collected supernatant was subsequently mixed with an equal amount of Heparin-agarose (Sigma-Aldrich) and kept at 4 °C overnight with constant agitation. The following day, the agarose-virus mixture was transferred to a chromatography column and the agarose was allowed to settle. The supernatant was then drained from the column by means of gravity and the agarose-bound virus was washed once with PBS and then eluted using PBS supplemented with 0.5 M NaCl. The eluted virus was then filtered again, desalinated, and concentrated using a 100 kDa centrifugal filter and then aliquoted and stored at −80 °C until use. For a full list of AAV plasmids used in this study, see Supplementary Table 2.

### Animals

All transgenic mouse lines used in this study have been previously characterized (Supplementary Table 3). In all experiments, male and female mice were used interchangeably in equal amounts, in an age

range, which varied between 1 and 6 months. In the behavioral experiments, slightly larger effects of TMP in females were noted, likely due to a more robust effect of TMP, but this observation was not further pursued. Experiments on C57BL/6 wild-type and transgenic mice were performed in strict accordance with institutional, national, and European guidelines for animal experimentation and were approved by the Bundesministerium für Wissenschaft, Forschung und Wirtschaft and Bildung, Wissenschaft und Forschung, respectively, of Austria (A. Haslinger, Vienna; BMWF-66.018/0010-WF/V/3b/2015; BMBWF-66.018/0008-WF/V/3b/2018).

## Stereotaxic intracranial virus injections

For in vivo delivery of viral vectors, mice were anesthetized with isoflurane, injected with buprenorphine 0.1 mg kg$^{-1}$, and placed in a stereotaxic frame where they continued to receive 1–5% isoflurane vaporized in oxygen at a fixed flow rate of 1 l min$^{-1}$. Leg withdrawal reflexes were tested to evaluate the depth of anesthesia and when no reflex was observed an incision was made across the scalp to expose the skull. Bregma was then located and its coordinates used as reference for anterior-posterior (AP) and medio-lateral (ML) coordinates, while the surface of the dura at the injection site was used as reference for dorso-ventral (DV) coordinates. In our experiments, we used the following sets of AP/ML/DV coordinates (in mm): DG: −1.9/1.3/−1.9; CA3: −1.9/±2.5/−2; CA1: −1.9/1.5/−1.2; hippocampal INs: −1.9/1.8/−1.6; EC: −4/3.5/−3. AAV vectors were first diluted 1:5 in PBS and delivered to the injection site at a volume of 0.3 μl and a rate of 0.06 μl per minute, using a Hamilton syringe and a 32G needle. After the injection was completed, the needle was left in place for an additional 1–2 min to allow the virus to diffuse in the tissue, and then slowly retracted. At the end of the injection session, the scalp was glued back and the mice were returned to their home cage to recover. Injection of pseudotyped rabies viral vectors took place 2–3 weeks after initial injection of AAV vectors containing the TVA receptor and rabies glycoprotein. Pseudotyped rabies vectors were first diluted to reach a final concentration of ~2–5 × 10$^8$ TU ml$^{-1}$ and then injected in the same manner as the AAV. Except for rabies injections into the DG of Prox1-cre or EC of Ctgf-2A-dgCre transgenic animals, all other injections of rabies virus were shifted −0.2 mm AP and −0.2 mm ML. This was done in order to avoid, as much as possible, non-specific labeling along the needle tract of the first injection, due to the lack of complete specificity of Cre-recombinase expression in the KA1-cre and DLX5/6-Flp transgenic lines.

## Slice preparation and electrophysiology

Electrophysiological recordings from identified retrogradely-labeled cells were performed 5–7 days following injection of CVS-N2c vectors. Manipulated animals were anesthetized using an MMF mixture consisting of medetomidin (0.5 mg kg$^{-1}$), midazolam (5 mg kg$^{-1}$) and fentanyl (0.05 mg kg$^{-1}$) and transcardially perfused with 20 ml ice-cold dissection solution containing 87 mM NaCl, 25 mM NaHCO$_3$, 2.5 mM KCl, 1.25 mM NaH$_2$PO$_4$, 10 mM D-glucose, 75 mM sucrose, 0.5 mM CaCl$_2$, and 7 mM MgCl$_2$ (pH 7.4 in 95% O$_2$/5% CO$_2$, 325–327 mOsm). The brain was then removed and the hippocampus along with the adjacent cortical tissue was dissected out and placed into a precast mold made of 4% agarose designed to stabilize the tissue. The mold was transferred to the chamber of a VT1200 vibratome (Leica Microsystems) and the tissue was transversely sectioned into 350-μm-thick slices in the presence of ice-cold sectioning solution. Transverse corticohippocampal slices were allowed to recover for ~30 min at ~31 °C and then kept at room temperature (RT, 22 ± 1 °C) for the duration of the experiments. During recordings, slices were superfused with recording solution containing 125 mM NaCl, 2.5 mM KCl, 25 mM NaHCO$_3$, 1.25 mM NaH$_2$PO$_4$, 25 mM D-glucose, 2 mM CaCl$_2$, and 1 mM MgCl$_2$ (pH 7.4 in 95% O$_2$/5% CO$_2$, 316 mOsm), at a rate of ~1 ml min$^{-1}$ using gravity flow. Neurons in the regions of interest were patched using pulled

patch pipettes containing: 125 mM K-gluconate, 20 mM KCl, 0.1 mM EGTA, 10 mM phosphocreatine, 2 mM MgCl$_2$, 2 mM Na$_2$ATP, 0.4 mM Na$_2$GTP, 10 mM HEPES (pH adjusted to 7.28 with KOH, ~300 mOsm); 0.3% biocytin was added in a subset of recordings. Patched cells were maintained in current-clamp mode at the neuron's resting membrane potential. Signals from patched cells were acquired using an Axon Axopatch 200 A amplifier (Molecular Devices) and digitized using a CED Power 1401 analog-to-digital converter (Cambridge Electronic Design). Optogenetic stimulation was delivered using a blue-filtered white LED at an intensity of 4.5 mW mm$^{-2}$ (Prizmatix, IL), passed through a ×63 objective positioned above the recorded neuron. For all recordings, 2–5 light pulses of 5 ms duration were delivered at a frequency of 10 Hz, with 20 s interval between stimulations. For isolation of AMPA and NMDA receptor-mediated currents, we used an internal CsCl-based solution comprised of 145 mM CsCl, 10 mM HEPES, 2 mM MgCl$_2$, 5 mM phosphocreatine 2 mM Na$_2$ATP, 0.3 mM Na$_2$GTP, and 5 mM QX-314 (pH adjusted to 7.28 with KOH, ~310 mOsm), along with the following pharmacological agents: 100 μM Gabazine, 25 μM CNQX and 50 μM D-AP5 applied at different stages of the recording. At the end of each recording session, the electrode was slowly retracted to form an outside-out patch. Subsequently, the slice was removed from the recording chamber, submerged in 4% paraformaldehyde (PFA) and then kept in 0.1 M phosphate buffer (PB) until further processing. Subsequent analyses of EPSC and membrane properties were performed using Stimfit (version 0.15.8; https://github.com/neurodroid/stimfit)[74] and representative traces were processed and visualized using Igor Pro 6 (Wavemetrics, OR, USA).

## Fixed tissue preparation and immunolabeling

For imaging purposes, animals were sacrificed 5–7 days after injection of RVdG$_{envA}$-CVS-N2c vectors. First, animals were anesthetized as described in the previous section and transcardially perfused with 15 ml of 0.1 M PB followed by 30 ml of 4% PFA. Following perfusion, the brain was removed and kept in 4% PFA overnight at 4 °C, which was subsequently replaced with 0.1 M PB. Fixed brains were sectioned 100-μm thick in either a coronal, parasagittal or transverse plane and stored in 0.1 M PB at 4 °C.

For immunohistochemical labeling of transduced tissue, standard protocols were used. First, the sections were washed with PB 3 times for 10 min. Next, sections were incubated with 10% normal goat serum (NGS) and 0.3% Triton X-100 for 1 h, at RT with constant agitation and subsequently with rabbit antibodies against CPLX3 (Synaptic Systems, Cat#122 302, 1:500), in PB containing 5% NGS and 0.3% Triton X-100, at 4 °C overnight. After washing, slices were incubated with isotype-specific secondary antibodies (goat anti-rabbit conjugated with Alexa Fluor 488 or 647) in PB containing 5% NGS and 0.3% Triton X-100 for 2 h in RT with constant agitation. After washing once more, slices were mounted, embedded in Prolong Gold Antifade mountant (Thermo-Fisher Scientific, Cat# P36930) sealed with a 0.17-mm coverslip. For double labeling of CPLX3 with VGLUT1 or VGAT, guinea-pig antibodies were used (Synaptic Systems, Cat# 135304 and 131004, respectively, 1:200) and for labeling of PCP4 and NeuN, rabbit antibodies were used (Novus Biologicals, Cat# NBP1-80929 and NBP1-92693 respectively, 1:500). For acquisition of STED images, we used anti-rabbit Star580 and anti-guinea pig Star635P secondary antibodies (Abberior, Germany, 1:200).

Neurons that were filled with biocytin (0.3%) were processed for morphological analysis. After withdrawal of the pipettes, resulting in the formation of outside-out patches at the pipette tips, slices were fixed for 12–24 h at 4 °C in a 0.1 M PB solution containing 4% PFA. For reconstruction of dendritic morphology, fixed slices were treated with Alexa-Fluor 647-conjugated strepdavidin (Invitrogen, Cat# S32357: 1:200) along with 5% NGS and 0.4% Triton X-100 for ~2 h followed by 3 wash cycles in PB. Stained sections were then mounted on a slide, embedded in Mowiol (Sigma-Aldrich) and sealed with a 0.17-mm coverslip.

## Fluorescent in situ hybridization (FISH)

An adult wild-type mouse (P60) was euthanized by decapitation and its brain was quickly removed, fixed in cold 4% PFA for 4 h, followed by an additional 24-h incubation in a cold 30% sucrose solution. Next, the brain was embedded in optimal cutting temperature compound (OCT, Tissue-Tek), frozen with dry ice, and sliced at 16 μm sections with a cryostat (Leica CM1950, Leica Biosystems). First, sections were fixated with 4% PFA for 15 min, washed with PBS, and dehydrated using an ascending EtOH gradient (25%, 50%, 75%, and 100%, each step for 5 min with subsequent drying for 15 min). The slices were then stained according to the manufacturer's protocol[75] (Molecular Instruments), with DAPI supplemented prior to mounting. ISH probes for detection of *Cplx3* and *Ctgf* (also termed *Ccn2*) were commercially designed by the manufacturer.

## Microscopy

Confocal images were acquired using an LSM 800 microscope (Zeiss). A subset of cleared tissue samples was acquired using Andor Dragonfly spinning disk confocal microscope (Andor technologies). All representative confocal images displayed in this manuscript are shown as a maximal intensity projection of an image stack of 4–12 separate images. For imaging of intact tissue, the cortical plate, containing the hippocampus and adjacent EC, was removed from the transduced hemisphere and subsequently cleared using the CUBIC method[76]. Once the tissue appeared translucent, it was mounted into a chamber still immersed in the CUBIC solution, slightly pressed in order to flatten the tissue, and topped with a coverslip. The cleared tissue was then transferred to the confocal microscope with the CA3 side facing up, where it was imaged overnight. For all images used in this manuscript, possible cross-talk between channels was closely monitored and was ruled out prior to analysis of signal overlap.

Dual-color STED microscopy was performed on a commercial inverted STED microscope (Abberior Instruments, Germany) with pulsed excitation and STED lasers. A 561 nm and a 640 nm laser were used for excitation and a 775 nm laser for stimulated emission depletion. An oil immersion objective with numerical aperture 1.4 (UPLSAPO 100XO, Olympus, Japan) was used for image acquisition. The fluorescence signal was collected in a confocal arrangement with a pinhole size of 0.6 airy units using photon counting avalanche photodiodes with 605/50 nm and 685/70 nm bandpass filters for STAR 580 or STAR 635P detection, respectively. The pulse repetition rate was 40 MHz and fluorescence detection was time-gated. The imaging parameters used for acquiring dual-color STED images were 15 μs pixel dwell time, ~4.5 μW (561 nm) and ~3.8 μW (640 nm) excitation laser power, and ~75 mW STED laser power. The channels were acquired consecutively as line steps with two-line accumulations for each channel. For 3-color images, the same region of interest was recorded with a third color channel with diffraction-limited resolution using a 488 nm laser with 25-μs dwell time and ~40 μW excitation power. Here, a pinhole size of 1.0 airy unit and 2-line accumulations were used. The signal was collected using a photon counting avalanche photodiode with a 525/50 nm bandpass filter. Pixel size was 40 nm for all images. The power values refer to the power at the back aperture of the objective. Images from the hippocampal *S.M.* were taken from the regions corresponding to the projections from the MEC, i.e., the medial molecular layer of the DG, CA3, and CA2, the proximal molecular layer of the CA1, and the distal *S.M.* of the subiculum.

## Image analysis

Quantification of cell numbers in the cleared tissue preparation, as well as quantification of colocalization in thin sections, was performed using Imaris software (Oxford Instruments) with a maximal distance for colocalization set at 2 μm. Analysis of terminal colocalization for STED images was performed using Fiji ("Fiji is just ImageJ")[77] via a custom-written script (https://github.com/sommerc/coloco3surf). For

this purpose, a mask was created for each of the individual channels following manual adjustment of the threshold, with surfaces smaller than 0.1 μm² discarded. The signal density for each channel was then calculated as the ratio between the total number of pixels in the mask and the total number of pixels in the image. Next, a new mask, which contained only overlapping pixels from the two channels was created, with surfaces smaller than 0.1 μm² again discarded and the density calculated as described for the primary masks.

## Microdrive implantation

Ctgf-2A-dgCre//Ai40 mice were administered with TMP 150 mg kg⁻¹ in order to drive the expression of ArchT-GFP in EC-6b neurons. 2–3 days after Cre-induction, animals were implanted with 6 independently movable tetrodes, surrounding an optic fiber (240 μm core diameter, 0.63 NA, 45° conical tip; Doric Lenses), under deep anesthesia using isoflurane (0.5–3%), oxygen (1–2 l min⁻¹), and an initial dose of buprenorphine (0.1 mg kg⁻¹). The tetrodes were constructed from four 12-μm tungsten wires correspondingly (H-Formvar insulation with Butyral bond coat, California Fine Wire, Grover Beach CA), twisted, and then heated in order to bind them into a single bundle. The tips were then gold-plated to reduce their electrode impedance to 200–400 kΩ. During surgery, a craniotomy was centered above the dorsal CA1 and the tip of the optic fiber was implanted into the following coordinates: −1.9 mm AP and 1.6 mm ML from bregma and −1.2 mm from the pial surface. The tips of the tetrodes were initially positioned ~300 μm above the tip of the fiber. Two screws positioned above the cerebellum served as ground and reference electrodes. Two additional stainless-steel anchor screws were used to permanently attach the microdrive assembly to the skull. The paraffin wax-coated electrodes and the microdrive apparatus were then daubed with dental acrylic to encase the electrode-microdrive assembly and anchor it to the screws in the skull. Following a recovery period of 7 days, the tetrodes were lowered in 50–150 μm steps each day into the CA1 region over a further period of 7–14 days.

## Single-unit data acquisition and analysis

Extracellular data was acquired using the Intan RHD2000 evaluation system and a RHD2132 headstage (Intan Technologies, CA, USA) together with an extended version of the Ktan software (https://git.ist.ac.at/alois.schloegl/ktan.git). The electric signal was sampled at 20 kHz. Trajectory was tracked with a video camera (FL-HC0614-2M; RICOH) recording two LED lights attached to the headstage and recorded using the Positrack software (https://github.com/kevin-allen/positrack). Spike extraction was performed by implementing the Mountainsort software[78] and a customized software was used for cluster cleaning and refinement (https://github.com/igridchyn/lfp_online). Green light for ArchT activation was provided by a 535 nm Green LED at 2.5 mW mm⁻² (Prizmatix, IL). The light source was coupled into a 0.48 NA optic fiber patch cord (4.5 m long, 0.37 NA; Doric Lenses), which transmitted the light to the microdrive. The light pulses were triggered with TTL pulses sent through the parallel port of the computer, running the acquisition and real-time signal decoding software.

For data analysis, the open field environment was divided into $15 \times 15$ equally sized spatial bins of approximately 11.1 cm² each. Using the positional tracking, an occupancy map was generated by computing the amount of time the animal spent in each spatial bin during running periods (speed filter >3 cm s⁻¹). Then the number of spikes a cell fired in each spatial bin were counted (also speed filtered, >3 cm s⁻¹) and divided by the occupancy time. Rate maps were then smoothed with a Gaussian filter with a standard deviation of 1 bin. To measure the spatial tuning of cells, the SI measure was calculated[79]. Only cells with a mean firing rate between 0.2 and 5 Hz were included in this analysis, thereby constituting putative pyramidal cells. Furthermore, cells had to have a firing rate of at least 0.2 Hz in both the first and last

15 minutes of the recording to ensure inclusion of only stably recorded cells. For each cell, the rate map was divided into four equally sized quadrants. The SI was separately calculated for each quadrant and the average SI between the light and non-light quadrants was compared. We used the following definition:

$$\text{Spatial information (SI)} = \frac{1}{\Sigma(P_{\text{bins}})} \sum_{x=1}^{\text{bins}} \left( \frac{\lambda_x}{\lambda} \log_2 \frac{\lambda_x}{\lambda} \right) \qquad (1)$$

where $\lambda_x$ is the mean firing rate in spatial bin $x$, $\lambda$ the mean firing rate over the entire environment and $P$ the occupancy[79].

### Intellicage experiments

For assessment of the contribution of EC-6b neurons to spatial memory, we crossed Ctgf-2A-dgCre and RosaStopDTA mice to enable conditional ablation of the subplate layer. RosaStopDTA mice are thought to induce highly specific cell ablation, because mice lack a functional receptor for diphtheria toxin and DTA lacks the B subunit required for penetration of cell membranes[80]. Double transgenic male and female offspring 2–5 months old, were implanted subcutaneously with an identifier transponder and 1 week later placed in an intellicage environment (TSE-Systems, Germany)—up to 8 mice simultaneously of the same gender, with a constant temperature, a 12-h light cycle and ad libitum access to food. Upon entry to the intellicage, the animals underwent a 5-day habituation period, during which all water ports were available and gradually learned how to access the water bottle following a 10-s delay from the moment of the nosepoke into the port. Next, each animal was randomly assigned to an individual water port, out of the eight available in each cage, for a period of 5 days. At the end of this initial training period, the animals were randomly divided into the experimental group, which were administered i.p. with 150 mg g$^{-1}$ TMP, and a control group injected with a vehicle. Individual error rate was measured as the ratio between nosepokes to the assigned port and total number of nosepokes during a 24-h period. Mice that failed to reach an error rate lower than 70% by the last day of the initial learning period were excluded from the study and all further analyses. Immediately following this manipulation, the mice were placed back in their intellicage and assigned with a different water port, for a period of 3 days. After three such rotations, all consisting of identical changes to both corner and side, the animals were assigned again to their original water port, which was allocated to them before the manipulation, for an additional 5-day period. At the end of this trial, the animals were removed from the cage, transcardially perfused with PFA 4% in PB, and the brains from the experimental group sectioned at 100 μm for further analysis of CPLX3 synaptic density. Apart from the administration of TMP or vehicle, no direct or indirect contact was made with the animals.

### Statistics and reproducibility

All values were reported as mean and error bars as ± SEM. Statistical significance was tested using a non-parametric, single-sided Kruskal–Wallis test followed by a double-sided Mann–Whitney test for post hoc comparisons, or by Fisher's exact test for analysis of differences in group proportions. Multiple comparisons were adjusted using Holm–Bonferroni correction. Confocal images used as representation were replicated successfully for over different animals, with identical results. For analysis of intellicage data, we used a single-sided non-parametric aligned rank transform three-way ANOVA test to compare sessions A, D, and A*[81] and a single-sided non-parametric Friedman's test to examine the learning time course. Calculations were performed in Microsoft Excel, Python, or R (version 4.1.0). Statistical differences with $p < 0.05$ were considered significant. In figures, a single asterisk (∗), double asterisks (**), and triple asterisks (***) indicate $p < 0.05$, $p < 0.01$, and $p < 0.001$, respectively, and are used throughout the manuscript.

### Reporting summary

Further information on research design is available in the Nature Research Reporting Summary linked to this article.

### Data availability

Additional image data files are available from the corresponding author upon reasonable request. Source data are provided with this paper.

### Code availability

The following custom code was generated for this study and is available through the following links: Coloco3surf–https://github.com/sommerc/coloco3surf. Ktan–https://git.ist.ac.at/alois.schloegl/ktan.git. Analysis routines are available from the corresponding author upon reasonable request.

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

## Acknowledgements

We thank F. Marr and A. Schlögl for technical assistance, E. Kralli-Beller for manuscript editing, as well as C. Sommer and the Imaging and Optics Facility of the Institute of Science and Technology Austria (ISTA) for image analysis scripts and microscopy support. We extend our gratitude to J. Wallenschus and D. Rangel Guerrero for technical assistance acquiring single-unit data and I. Gridchyn for help with single-unit clustering. Finally, we also thank B. Suter for discussions, A. Saunders, M. Jösch, and H. Monyer for critically reading earlier versions of the manuscript, C. Petersen for sharing clearing protocols, and the Scientific Service Units of ISTA for efficient support. This project was funded by the European Research Council (ERC) under the European Union's Horizon 2020 research and innovation programme (ERC advanced grant No 692692 to P.J.) and the Fond zur Förderung der Wissenschaftlichen Forschung (Z 312-B27, Wittgenstein award for P.J. and I3600-B27 for J.G.D. and P.V.).

## Author contributions

Y.B. conceived the project, designed and performed all experiments, Y.B. and P.V. acquired STED images advised by J.G.D., K.K., and Y.B. analyzed single-unit data, J.C. provided facility and equipment for tetrode recordings, Y.B. analyzed the data, Y.B. and P.J. wrote the manuscript, and P.J. supervised the project. All authors read and commented on the manuscript.

## Competing interests

The authors declare no competing interests.
