## [Peer Review File · Nature Communications]

A direct excitatory projection from entorhinal layer 6b neurons to the hippocampus contributes to spatial coding and memoryEditorial Note: This manuscript has been previously reviewed at another journal that is not operating a transparent peer review scheme. This document only contains reviewer comments and rebuttal letters for versions considered at *Nature Communications*.

REVIEWER COMMENTS

Reviewer #1 (Remarks to the Author):

After rereading the revised manuscript multiple times, I arrived at the conclusion that the advances presented in the study outweigh the limitations of the approach. I also commend the authors for being transparent and forthcoming about these current technical limitations in their revised discussion. I support the publication of the manuscript in *Nature Communications*.

Reviewer #4 (Remarks to the Author):

The manuscript by Ben-Simon et al. uses state-of-the art anatomical methods to characterise a largely ignored connection from entorhinal L6b cells to the hippocampus. There is currently no published literature on the function of L6b anywhere in the adult cortex, so this paper is highly novel in its claims that the EC-L6b neurons contribute to hippocampal spatial coding, and that disrupting this circuitry with diphtheria toxin ablation impairs spatial memory.

The anatomical work presented is detailed and of good quality, although the write-up is occasionally careless (e.g. p8 'Cplx3+ EC-1' is not previously specified, and probably should refer to Cplx3+ EC-5). I don't quite understand why the authors refer to the cells as L6b SPNs, that appears an unnecessary complication. They use adult brains to describe the circuitry, so should be referring to these cells as L6b cells, not subplate neurons.

There are attempts in the text to link the EC-L6b Cplx3+ or Ctgf-dgCre+ neurons to the developmental subplate. This is not necessary for this study to be impactful, and is a relatively weak link, as the references they cite have not studied gene expression in entorhinal cortex. As many developmental subplate markers are also expressed in other brain regions, at least during development, the authors really ought to determine the birth-date of their EC deep layer cells if they want to make the link to the developmental, early-born subplate (note, they would need to show that the EC-L6b cells are amongst the earliest born neurons in that region, not that they are born on E11.5 or a similarly arbitrary time). Alternatively they can just refer to these as L6b cells. In the discussion, if they want to make the link to the early-born subplate, and the preferential survival of Cplx3+ and Ctgf+ SPNs, they should cite Hoerder-Suabedissen & Molnar 2013, which actually shows this for S1, not Hoerder-Suabedissen et al., 2013 (which does so only indirectly).

The authors demonstrate a striking behavioural effect of both optogenetic silencing of Ctgf-dgCre+ neurons and ablation of the same. While their optogenetic light manipulation is restricted to the hippocampus, the source of the manipulated fibres is not unequivocally established as coming from EC-L6b. Ctgf-dgCre+ cells are present throughout L6b of the cortex. If the authors want to make the claim that these are EC-located L6b cells, they will need to show evidence from Rabies tracing that there are no Cplx3+/Ctgf-dgCre+ cells in any other forebrain region with a monosynaptic connection to the hippocampus. And – yes – I'm aware that Rabies tracing is not infallible and can fail to show existing connections, but it is still better than the current state of showing no evidence either way.

Similarly, the DTA ablation of Ctgf-dgCre+ cells is a cortex-wide manipulation. The authors argue that an EC-restricted approach is not feasible. But again, the authors need to show that there are no other Cplx3+ L6b or L5 cells from other brain regions contributing to hippocampal eCplx3+ terminals if they persist in referring to these as 'EC-6b' cells. The effect of Ctgf-dgCre+ cell ablation is striking – why

not just refer to it as generally L6b, without insisting that they must come from EC, if other sources have not been ruled out?

I'm not sure that the authors can rule out neuroinflammation as a contributing source to the behaviour phenotype after DTA. There are no controls to show that synaptic degeneration in another pathway (e.g. EC2 input) in the CA3 region does not produce a behavioural effect.

The authors should include some discussion on why they believe the loss of EC-L6b Cplx3+ input correlates strongly with the impaired performance on day 8, but not on day 9, if the 'impaired forgetting' on day 9 is also ascribed to the loss of EC-L6b cells.

Minor comments: the authors mention that EC-L6b SPNs might be ideally placed to synchronise network activity. However, in Fig 4, optogenetic stimulation of these cells appears to result in poorly synchronised CA3 neuron firing, in contrast to the tightly timed AP generation after perforant path or mossy fibre optogenetic stimulation. The authors should comment on this as part of the discussion, provide more evidence for why they think the role might be a synchronising one, or remove the suggestion that these cells play a synchronising role.

The authors should show some higher magnification images of panel Fig 5d. Previously they report the presence of sparse Cplx3+ INs in the hippocampus, as well as showing that these do not co-localise with the Ctgf-dgCre+ cells, yet their low magnification images in Fig 5d make it seem as if all Cplx3 immunofluorescence has disappeared from the hippocampus after DTA.

Extended Figure 1 – resolution of panels d+e is very poor – please improve if that isn't just a flaw of the pdf generated for reviewers. Typo in legend (is-situ should read in situ).

Extended Figure 2 – correct the figure legend to remove Allen Brain Atlas acknowledgement for panels a & b.

Figure 2 – the figure panels should include labels/colour description with the panel for Fig 2b, to make it clear which colour represents what.

Extended Figure 5 – refers to L5 Cplx3+ cells as SPN, please correct this. Panels b-d require a layer and/or region annotation for the lower magnification images.

Extended Figure 8 – panels a+b: colours need to be explained in the panel. Typo in legend (mise should be mice).

Figure 4 and extended Figure 9: both claim to state number of repeats in legend, but neither does.

Point-by-point reply to reviewer's comments

Reviewer #1 (Remarks to the Author):

After rereading the revised manuscript multiple times, I arrived at the conclusion that the advances presented in the study outweigh the limitations of the approach. I also commend the authors for being transparent and forthcoming about these current technical limitations in their revised discussion. I support the publication of the manuscript in Nature Communications.

We would like to thank the reviewer for the extra time spend on our manuscript and his / her favorable opinion. We hope that we will be able to address technical limitations and new questions in future work.

Reviewer #4 (Remarks to the Author):

The manuscript by Ben-Simon et al. uses state-of-the art anatomical methods to characterise a largely ignored connection from entorhinal L6b cells to the hippocampus. There is currently no published literature on the function of L6b anywhere in the adult cortex, so this paper is highly novel in its claims that the EC-L6b neurons contribute to hippocampal spatial coding, and that disrupting this circuitry with diphtheria toxin ablation impairs spatial memory.

The anatomical work presented is detailed and of good quality, although the write-up is occasionally careless (e.g. p8 'Cplx3+ EC-1' is not previously specified, and probably should refer to Cplx3+ EC-5).

We thank the reviewer for his / her positive comments ("highly novel", "detailed", "good quality"). We have double-checked acronyms and added explanations. However, in the example given by the reviewer, the original statement is correct, as we refer there to CPLX3+ layer 1 interneurons, and not to the layer 5 excitatory neurons. We have modified the text so that this distinction is made clearer (p. 8 of the revised manuscript).

I don't quite understand why the authors refer to the cells as L6b SPNs, that appears an unnecessary complication. They use adult brains to describe the circuitry, so should be referring to these cells as L6b cells, not subplate neurons. There are attempts in the text to link the EC-L6b Cplx3+ or Ctgf-dgCre+ neurons to the developmental subplate. This is not necessary for this study to be impactful, and is a relatively weak link, as the references they cite have not studied gene expression in entorhinal cortex. As many developmental subplate markers are also expressed in other brain regions, at least during development, the authors really ought to determine the birth-date of their EC deep layer cells if they want to make the link to the developmental, early-born subplate (note, they would need to show that the EC-L6b cells are amongst the earliest born neurons in that region, not that they are born on E11.5 or a similarly arbitrary time). Alternatively they can just refer to these as L6b cells. In the discussion, if they want to make the link to the early-born subplate, and the preferential survival of Cplx3+ and Ctgf+ SPNs, they should cite Hoerder-Suabedissen & Molnar 2013, which actually shows this for S1, not Hoerder-Suabedissen et al., 2013 (which does so only indirectly).

To address the concern of the reviewer, we have completely eliminated any reference to subplate neurons from the Results section. We now refer to these cells as EC-6b neurons throughout the

manuscript. We remain convinced that reference to the possible identity of the layer 6b neurons and subplate neurons in the Discussion section will be very useful for the readership. Our results suggest a new relation between brain development and mature network function, which can be further tested in future experiments. We have therefore left the corresponding Discussion paragraph in the manuscript. We now cite Hoerder-Suabedissen & Molnar 2013 (now Ref. 22) as requested (p. 14, bottom, of the revised manuscript).

The authors demonstrate a striking behavioural effect of both optogenetic silencing of Ctgf-dgCre+ neurons and ablation of the same. While their optogenetic light manipulation is restricted to the hippocampus, the source of the manipulated fibres is not unequivocally established as coming from EC-L6b. Ctgf-dgCre+ cells are present throughout L6b of the cortex. If the authors want to make the claim that these are EC-located L6b cells, they will need to show evidence from Rabies tracing that there are no Cplx3+/Ctgf-dgCre+ cells in any other forebrain region with a monosynaptic connection to the hippocampus. And – yes – I’m aware that Rabies tracing is not infallible and can fail to show existing connections, but it is still better than the current state of showing no evidence either way.

Similarly, the DTA ablation of Ctgf-dgCre+ cells is a cortex-wide manipulation. The authors argue that an EC-restricted approach is not feasible. But again, the authors need to show that there are no other Cplx3+ L6b or L5 cells from other brain regions contributing to hippocampal eCplx3+ terminals if they persist in referring to these as ‘EC-6b’ cells. The effect of Ctgf-dgCre+ cell ablation is striking – why not just refer to it as generally L6b, without insisting that they must come from EC, if other sources have not been ruled out?

We thank the reviewer for this comment. This important point is exactly addressed in ED Fig. 8, where we show that of all the inputs to the hippocampus, only cells in the entorhinal 6b colocalize with the reporter for the Ctgf-2A-dgCre line. Regions which do not appear in this plot did not send projections to the hippocampus. In the course of this study, we have performed numerous tracing experiments from hippocampal neurons and can confirm that this list is exhaustive. This result demonstrates that neurons in EC-6b are the only possible source of Ctgf+ input into the hippocampus and consequently, the only one affected by our experimental manipulations. We have added a sentence to the Results section to clarify this point (p. 11 bottom of the revised manuscript). Furthermore, we have added missing labels to ED Fig. 8 a and b, and reversed the y axis label of ED Fig. 8c ((tdTomato + EGFP+) / EGFP+) to improve clarity.

I’m not sure that the authors can rule out neuroinflammation as a contributing source to the behaviour phenotype after DTA. There are no controls to show that synaptic degeneration in another pathway (e.g. EC2 input) in the CA3 region does not produce a behavioural effect.

We thank the reviewer for pointing out this possibility. We consider this unlikely for several reasons. First, it is generally thought that cell ablation using the Rosa26-DTA mice is highly cell-specific. Specificity arises, presumably, because mice lack a functional receptor for diphtheria toxin and DTA lacks the B subunit required for penetration of cell membranes (Plummer et al., 2017, DOI: 10.1002/dvg.23067). Second, we could not detect any signs of neuroinflammation in our post-hoc morphological analysis, neither macroscopically nor microscopically (e.g. by immunolabeling). With the exception of the lack of cells in layer 6b, the brain sections appeared completely normal. Finally, VGlut1 synaptic density was

completely unchanged (Fig. 5d), showing that other inputs are unaffected by the manipulation. We have added a sentence to the Methods section to clarify these points, and also provide more details on the Rosa 26 Stop floxed DTA mouse line used (p. 28, bottom of the revised manuscript).

Plummer NW, Ungewitter EK, Smith KG, Yao HH, Jensen P. A new mouse line for cell ablation by diphtheria toxin subunit A controlled by a Cre-dependent FLEX switch. *Genesis*. 2017 Oct;55(10):10.1002/dvg.23067.

The authors should include some discussion on why they believe the loss of EC-L6b Cplx3+ input correlates strongly with the impaired performance on day 8, but not on day 9, if the 'impaired forgetting' on day 9 is also ascribed to the loss of EC-L6b cells.

The deficits both on day 8 and on day 9 correlate with the degree of layer 6b input loss, in opposite directions, consistent with our hypothesis. However, due to the stringent statistical approach we took (using the non-parametric Spearman test), only the result on day 8 is statistically significant while the result on day 9 only comes up as a non-significant trend (although it is significant with a parametric test). In any case, our results show that the slope of the regression line on both days goes in opposite directions, which means that there is an interaction between these two parameters, in a layer 6b neuron-dependent manner. We have changed the sign in the correlation value of day 8 to reflect the negative correlation in this test and have revised the Results section for additional clarification (p. 13, top of the revised manuscript).

Minor comments: the authors mention that EC-L6b SPNs might be ideally placed to synchronise network activity. However, in Fig 4, optogenetic stimulation of these cells appears to result in poorly synchronised CA3 neuron firing, in contrast to the tightly timed AP generation after perforant path or mossy fibre optogenetic stimulation. The authors should comment on this as part of the discussion, provide more evidence for why they think the role might be a synchronising one, or remove the suggestion that these cells play a synchronising role.

We have added a sentence on p. 15, describing why EPSCs with slower kinetics are potentially more suitable for synchronizing neuronal activity across regions at low frequencies (such as theta or delta). Interestingly, previous modeling work revealed that fast excitation is a surprisingly inefficient synchronization mechanism (e.g. van Vreeswijk et al., 1994 .doi: 10.1007/BF00961879). While we discuss this in brief in the discussion (p. 16 middle of the revised manuscript), this is clearly beyond the scope of the present study.

The authors should show some higher magnification images of panel Fig 5d. Previously they report the presence of sparse Cplx3+ INs in the hippocampus, as well as showing that these do not co-localise with the Ctgf-dgCre+ cells, yet their low magnification images in Fig 5d make it seem as if all Cplx3 immunofluorescence has disappeared from the hippocampus after DTA.

Representative STED images used to calculate eCPLX3 density have been added to Fig. 5d as requested. In addition, quantification of the VGluT1 synaptic density from both CA3 and CA1 has been included, to show that the overall excitatory input to these regions, which does not arise from EC-6b, has not been affected by the manipulation. iCPLX3 terminals are preserved but are difficult to detect in the low-

magnification images. However, they can be seen in STED images at higher magnification, consistent with our hypotheses.

Extended Figure 1 – resolution of panels d+e is very poor – please improve if that isn't just a flaw of the pdf generated for reviewers. Typo in legend (is-situ should read in situ).

Higher resolution images have been uploaded in order to allow for better illustration of the results. Furthermore, the typographical error has been corrected.

Extended Figure 2 – correct the figure legend to remove Allen Brain Atlas acknowledgement for panels a & b.

We apologize for the oversight. The unnecessary text has been removed from the figure legend as requested.

Figure 2 – the figure panels should include labels/colour description with the panel for Fig 2b, to make it clear which colour represents what.

The missing information regarding the color scheme has been added to the figure legend as requested.

Extended Figure 5 – refers to L5 Cplx3+ cells as SPN, please correct this.

The terminology in the figure legend has been corrected.

Panels b-d require a layer and/or region annotation for the lower magnification images.

Annotations of the different entorhinal cortical layers have been added to ED Fig. 5b–d as requested.

Extended Figure 8 – panels a+b: colours need to be explained in the panel. Typo in legend (mise should be mice).

A description of the color scheme has been added to ED Fig. 8a as requested.

Figure 4 and extended Figure 9: both claim to state number of repeats in legend, but neither does.

Number of repeats (N) has been added to both Fig. 4 and ED Fig. 9.

We hope that, after this second round of careful revisions, our manuscript can go to press without any further delay.

Once more, we thank the reviewers for helping us to improve our manuscript.